# The Value of "Values": A Case Study on the Design of Value-Inclusive Multimedia Content for the Menorah Artefact Collection at the Hecht Museum, Haifa, Israel

Srushti Goud [1,*] , Vincenzo Lombardo [1] , Tsvi Kuflik [2] and Alan Wecker [2]

1 Dipartimento di Informatica, Università di Torino, 10149 Torino, Italy
2 Information Systems Department, Faculty of Social Sciences, University of Haifa, Haifa 3498838, Israel
* Correspondence: srushti.goud@unito.it

**Abstract:** Cultural heritage (CH) values are important for understanding the significance of heritage assets. For that reason, the presentation of CH should go beyond providing factual information. It should reflect relevant values that are held by the curators, the heritage experts and the communities of non-expert citizen stakeholders. Associating values with the information content in a CH communication product is a challenging task. Digital technologies require special attention to communicate values along with heritage information to achieve meaningful and impactful communication. In this paper, we focus on a socio-technological framework for the integration of values into the information content provided to visitors. We have designed, applied and evaluated an eight-stage process for the inclusion of CH values in the information content and their communication to museum visitors. It has been applied at the Hecht Museum, located at the University of Haifa, Israel, where museum artefacts are currently presented to the visitors with informational panels that have been designed without any attention to values. Two digital applications, built by applying the eight-stage process, were developed. One was designed to cover the information and heritage values already available within the museum descriptions. The other was developed by following the suggested process, which accounts for values that were collected through a review of the literature, interviews with experts and interactions with non-experts. The two applications were tested, iterated and evaluated to assess the impact of value inclusion. Results show that both visitors and experts appreciated the value-enhanced communication. The evaluation of user feedback has further substantiated the creation of content that is inclusive of CH values, for the communication of museum artefacts.

**Keywords:** cultural heritage communication; cultural heritage values; digital application; socio-technological framework

## 1. Introduction

The definition of the terms 'Cultural Heritage' and 'Heritage Values' has been shaped, contested and reshaped over the last century. This shows the need for a comprehensive understanding of these terms. Cultural Heritage (CH) encompasses tangible and intangible expressions of communities and their ways of life passed on from generation to generation. CH includes community customs, practices, places, objects, artistic expressions, artefacts, monuments, groups of buildings and sites that are deemed significant. They convey diverse values which are classified as symbolic, historic, artistic, aesthetic, ethnological or anthropological, scientific and social values [1]. Values of cultural heritage are defined as a set of characteristics perceived in heritage by certain individuals or groups [2] and different individuals or groups may have different typologies and ranges of values [3].

The communication of CH requires historically accurate information that is presented in a manner that appeals to the user. Novelty in technology and interface of a digital application has the capacity to draw their attention. For example, User Interface and User Experience (UI/UX) design can drive engagement through spectacularization [4]. The

presentation of applications from a UI/UX point of view has been widely surveyed and discussed previously [5,6]. However, while user engagement is a relevant aspect for the development of a CH communication project, its core is the content and the heritage values conveyed with the information.

Content used to create applications for the communication of CH requires detailed analysis and needs to stimulate the curiosity of the user while also delivering cultural knowledge. Studies of technologies used in the development of digital applications for the communication of heritage answer the question "How to present Cultural Heritage to 'users'?". The area of research which we are covering in this paper attempts to answer the question "What aspects of Cultural Heritage are being presented to 'users' and Why?". We approach the process of communication of CH with the view that including CH values in the content evokes user interest and is beneficial for cultural knowledge sharing especially with willing participants. To substantiate this view, we developed a creation and structuring process of content that intentionally included CH values to improve the impact of digital Cultural Heritage Communication (CHComm).

CH values being characteristics associated to heritage by different interested parties who may be groups or individuals, leads us into a situation where many different associations and meanings exist at the same time. Some of these associations may even contrast and potentially be in conflict with one another as we will see in the following sections. Even so, it is necessary to discover significant associations, verify their authenticity and ensure that the value associations, their meanings, implications and changes to these associations over time are faithfully communicated to the willing participant of a cultural knowledge sharing process. This paper suggests, implements and evaluates an eight-stage process for the inclusion of CH values and their communication to museum visitors using a web-based digital multimedia application. This socio-technological framework was demonstrated by creating a digital CHComm application that included CH values and was built for the Menorah collection of artefacts at the Hecht Museum in Haifa, Israel.

In Section 2, we discuss the position of values in heritage management and illustrate our approach within the state of the art in digital CHComm. Further in Section 3, we present the design and implementation of a web-based multimedia application based on an operational framework designed to include values in CHComm. In Section 4, we analyse and discuss the results of an open-ended questionnaire survey conducted for the multimedia application and conclude the paper in Section 5.

## 2. State of the Art—Values and Digital Communication of CH

In order to better illustrate our approach, we need to discuss the position of heritage values in CH management and also within the state of the art in digital CHComm. Research on the idea of values within CH contexts and its typologies has been going on since the late 19th century [7]. It is interesting to note here that the charters and guidelines including ICOMOS charters, UNESCO guidelines and other national and international mandates published by heritage authorities or professional associations, do not have any directives for the communication of heritage values. It has been observed that the communication of CH is treated as 'secondary' to conservation [8]. Even so, it has been said that "every act of heritage conservation—within all the world's cultural traditions—is by its nature a communicative act" [9]. To avoid any ambiguity, we would like to state here that the term 'Values' or CH values referred to here are purely the associations of values and meanings attached to heritage and not the organizational values or organizational culture as referred in other research [10] relating to CH institutions and their internal management of people.

### 2.1. Assessing Values for Conservation Planning and Communication

In a report titled 'Assessing the Values of Cultural Heritage' published by the Getty Conservation Institute, a provisional typology of heritage values for the conservation and management of heritage was suggested. This has been reproduced in Table 1. The split

between Sociocultural and Economic aspects of value was identified as problematic and the entire set of typologies was not meant to be exclusive or exhaustive [11].

**Table 1.** Provisional value typologies suggested by the Getty Conservation Institute for consideration in heritage conservation planning and management.

| Sociocultural Values | Economic Values |
| --- | --- |
| Historical | Use (market) |
| Cultural/Symbolic | Nonuse (nonmarket) |
| Social | Existence |
| Spiritual/Religious | Option |
| Aesthetic | Bequest |

A more recent re-examination of value typologies identified three stages for significance assessments with regards to CH [12]. These are listed below:

1.  Features of Significance (or) What is the heritage in question?
2.  Aspects of Value (or) Why is this heritage valuable?
3.  Qualifiers of Value (or) How valuable is it?

Both of these approaches show two aspects of CH value assessments. The first is the propensity to categorise values with typologies and the second is to assess the significance of heritage by understanding its value associations. Assigning value typologies is primarily a professional exercise guided by expert views and understanding. Assessments of significance can be an inclusive process that considers expert and non-expert perspectives (community views).

Communities play a central role in conserving and communicating both the heritage and its associated values from one generation to the next. Communities have systems in place to conserve and propagate heritage that they consider significant. For example, the Adat community in Bali, Indonesia, is historically responsible for the conservation of sites, such as temples, royal palaces and Balinese Hindu rituals of the Balinese Hindu population. A study carried out in Denpasar, Bali [13], recommended an integrated approach for the development of the region, which considers the CH values associated by the community to these sites, beyond considering solutions for the socio-economic and environmental issues. All stakeholders may have value associations with the CH and these stakeholders can be experts or non-experts, individuals or communities who see themselves as 'cultural insiders' and others (outsiders). These CH value associations can be significant, complementary or contrasting and may or may not be adequately recognized in the mainstream academic and/or popular discourse [14]. The complicated realities of cultural heritage conservation and communication demands the design of content for the communication of CH be done with the necessary nuance and authenticity that it deserves and as such places a huge responsibility on the shoulders of designers.

### 2.2. Digital Communication of CH and Collaborative Heritage Management

Digital communication has transformed every field of human endeavour, including CHComm. It has been suggested that CH could very soon include "technological contributions and information originating from all intelligences in network society" [15]. In an increasingly interconnected world, communities are able to generate, contribute and manage their heritage identity online. For example, the Chandernagore town in India has many buildings (bungalows) of the Indo-French colonial architecture style and most of these were in various states of disrepair, causing concern among the residents. A perceived apathy towards the heritage by the local government spurred the people into coordinated action. Multiple Facebook groups spawned sporadic preservation efforts on behalf of the local civil society, and this led to the creation of "the Heritage & People of Chandernagore project", in 2015, initiated by a privately funded architecture firm. The firm started a collaborative mapping project that identified the built, tangible heritage in the town, along

with the intangible heritage that "the local citizens perceived and valued as their heritage". The outcome of this effort was a 'web-home for the heritage of Chandernagore'. Interactive digital tools were used to update the website with heritage information collected by "citizen historians" (residents), who went door to door and captured the oral histories and memories in video or text formats. The website was promoted by blog posts and social media. The project received global attention when, in 2017, the Ambassador of France to India, H. E. M. Alexandre Ziegler visited Chandernagore and the website was used to cover the history of the town in an orientation [16].

### 2.3. Digital Communication of CH that accounts for Values

Studies on digital applications within museum environments have focused on technologies and design elements that improve the user experience. Authors have published detailed guides on designing user experience for various digital platforms such as websites, mobiles, wearable computing, social media and collaborative environments [17]. Digital technologies such as smart maps have been used to "present the 'story' of museum resources and knowledge as a journey". This 'Stories, Journey and Smart Maps' (SJSM) approach was shown to increase the emotional engagement of users with museum collections [18]. Literature reviews covering mobile, AR, VR and Mixed Reality technology in cultural heritage communication and tourism show the extent of technology adoption and data collection [5,6,19]. Improving user experience, increasing emotional engagement, encouraging technology adoption, enhancing digital presence and digital accessibility are necessary for heritage communication institutions such as museums in an increasingly inter-connected world with a growing number of digital-natives. It has been argued that "the large banks of objects and knowledge about our past held in museums are an extraordinary source of discovery, leisure and life-long learning for the emerging digital heritage tourist" [20]. While exceptional research is being carried out with regards to improving user experience and more research may be needed on the digital accessibility of museums, studies on the communication of CH that highlights the CH values inherent in the heritage properties and their intangible associations have not been frequent.

We take the view that the communication of CH can be enhanced by identifying, filtering and thoughtfully presenting CH values in digital content intended for willing participants. A noteworthy example of identifying CH values leading to effective conservation of community value associations can be seen in the implementation of the Mobile Museum project. The Mobile Museum project was built around the artefacts housed in Queensland museums to facilitate remote access for Nalik people of New Ireland, Papua New Guinea to reconnect with their heritage [21]. This project employed a participatory methodology to meet the needs of the community and was a collaborative exercise with the University of Queensland, the Queensland museum and the Nalik community. The designers were looking to provide digital access to wooden sculptures and carvings housed in the Queensland museum through their 3D recreations. These sculptures are an integral part of the Nalik communities funerary rites. According to the community traditions, these wooden sculptures carved after the death of a person and adorned with clan specific motifs are presented to the community with accompanying song and dance rituals at a 'mortuary feast'. Once the rituals associated with death were complete, these sculptures called '*malangan (also spelt as Malanggan or Malagan)*' were burnt or left to rot. These sculptures are then re-carved at the next mortuary feast by the specific carvers who carry out this task from memory. The CH value associated with the malangan is not in the physical sculpture and its carving but in the memories of the imagery and in the right to reproduce such carvings [22]. The Mobile Museum project designers were cognizant of these value associations and attempted to address concerns in the Nalik community that some clans had forgotten how to reproduce their malangan carvings. A digital platform was created through which members of the community could see the 3D recreations of malangan and even zoom in to understand the imagery and carvings on the sculptures. Textual descriptions and notes made by community members were also used to enhance

the digital recreation. Since only 10% of the community had reliable internet access on their mobile devices, the digital information was also circulated via CD ROMs to members of the community who could then view the information on their home computers. This helped members of the community reconnect with their heritage.

While this particular effort may appear to be a unique case to a casual external observer, the point of note here is that the value associations and meanings of the malangan which is also the name for the funerary rites practiced by the community, are important in understanding the value and significance of the artefacts housed in the Queensland museum. The conservation of the Nalik community values was executed well by the Mobile Museum project and the ideal next step that would enhance the conservation and propagation of the CH values uncovered here would be the communication of these Nalik community value associations to all interested visitors at the Queensland museum. We observe that research into this aspect of CHComm which encompasses the "What and Why" of the content being communicated especially within digital applications has been infrequent. Communication of CH that discusses the value associations, their meanings to the culturally connected communities or individuals and the changes to such meanings and associations over time is an essential aspect of conserving and managing CH. While the specifics of the value associations may vary from case to case as seen in the example of the Nalik community of New Ireland or the Adat Hindu community of Denpasar, the fact remains that the efforts of communication within CH needs to account for all significant, verifiable and authentic CH value associations.

An eight stage framework for the creation and structuring of content for a digital application was developed to account for CH values in the communication of museum artefacts. The implementation of this framework was done using a web-based multimedia application at the Hecht Museum in Haifa, Israel. This museum was the site of a previous project that aimed to create a 'visitor's guide to an active museum'. This application was designed to answer the various questions a visitor might have regarding a particular artefact and meant to be used as a guide during the museum visit. The approach here was to cater to a wide range of visitors and their potential questions with simplistic answers that were meant to be understood by those who may be viewing the artefact while simultaneously listening to the audio. The application was mobile-based and contained relevant illustrative images. The evaluation of the application was focused on the length of the visit, increase in knowledge about the artefacts, type of User Interface (adaptive versus non-adaptive UI) and user modelling mediation. The study found that users showed a significant increase in their learning about the artefacts and spent a longer time being guided through the exhibits when using the application [23]. The encouraging results of the study nevertheless did not account for CH values and perspectives associated with artefacts and intangible heritage concepts by different individuals and communities. Our approach to design and implement a digital multimedia application that intentionally included CH values in its content for the communication of selected artefacts within the Hecht museum and the evaluation of its effectiveness are discussed in the following sections.

## 3. Design and Implementation of Value-Inclusive Content for a Digital Multimedia Application

The deliberate and thoughtful inclusion of CH Values in CHComm applications and experiences is necessary to conserve and communicate heritage with its inherent meanings. The authenticity of the source of heritage information and values has to be verified to the best of their abilities by designers of such applications. It is not feasible to cover every source or type of value associated with a particular heritage property, when communicating about it. Even so, to instigate curiosity and facilitate an open discussion using a digital application, it should engage the user and provide them with new knowledge. Visitors coming to a museum are good candidates to be participants in the cultural knowledge sharing process. Therefore, the content and its delivery should "contribute to the overall visitor experience in a positive, enlightening, provocative, and meaningful way" [24], to a

"level of detail and the perspectives in which they are interested" [23]. Digital CHComm applications within museums sometimes face the challenge of having to link disparate objects that may not have obvious connections to each other. This could mean that some selected objects are included in a digital communication experience, while others are left out. Additionally, the constraints of the environment require designers to make such decisions. We have seen such efforts succeed with reasonably encouraging results [25]. Before we present our approach to the design and implementation of the application, it is imperative to briefly touch upon the background of the Hecht Museum at Haifa, Israel.

### 3.1. Background—Dr. Reuben Hecht and Eretz Israel

Dr. Reuben Hecht founded the Hecht Museum at Haifa university in 1984 to house his own collections of archaeological artefacts and 19th-century paintings. His vision was to showcase the connection between the people of Israel and 'Eretz Israel'. Eretz Israel is a Hebrew term that means 'Land of Israel' and he believed that archaeology could be an expression of Zionism. As such, the artefacts showcased in the museum cover the period of his interest which is from Canaanite era (2000 BCE) to the Roman/Byzantine era (500–550 CE) showing their influence over the region of modern-day Israel. In order to select an artefact or a group of artefacts for developing a digital multimedia application, we interacted with the museum staff. Based on their recommendations, four groups of artefacts were shortlisted as important and out of these groups, we selected the artefacts from the Menorah collection because of its role as the national emblem of Israel today. The content that was selected based on the recommendations are highlighted in Figure 1, with the other recommended artefacts also mentioned. The administrative importance associated with the Menorah symbol and the fact that many of these artefacts were daily use objects provided an opportunity to explore the contemporary perception of the symbol and the artefacts among the visitors. As the application was to be developed within the context of the museum, the content design included information about them and the CH values put forward by Dr. Reuben Hecht through his vision for the museum. An overview of Menorah as an artefact and a symbol was included in the content of the application before the users were provided with options to learn more about specific artefacts seen in the Menorah collection.

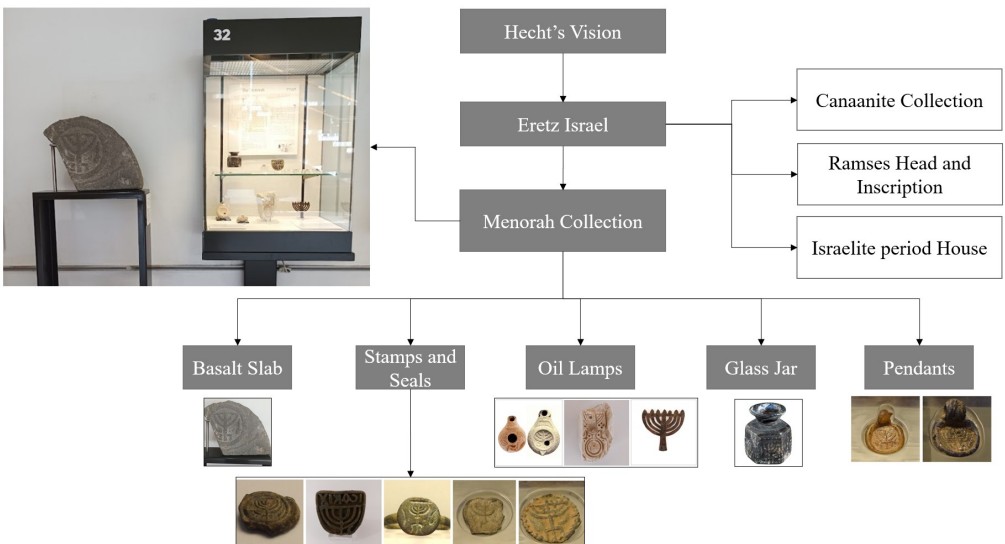

**Figure 1.** Recommended groups of artefacts in the Hecht Museum for the digital multimedia application with the final selection and its component sub-groups highlighted.

### 3.2. Design of Digital Multimedia Application for the Menorah Collection at the Hecht Museum

Effective inclusion and propagation of CH values in CHComm applications can be achieved by identifying the values to be shared, expressing them through the content

and then assessing its appreciation by users. The assessment of user appreciation can lead to an understanding of which CH values shared ended up making the most impact. Involving user and expert reviews in an early stage of design shows designers if they ignored any values that the community considers important. Consequently, evaluating the application leads to an improved iteration where the initial prototype is redesigned as necessary. This re-design can go from changing the words that are used in the content to adding, removing or changing aspects of the explanatory illustrations and visual elements in a digital multimedia application. This has been demonstrated to be beneficial in a project where users were asked for their requirements alongside experts prior to the design of the digital application [26]. Designers have also applied participatory approaches to include user expectations in Digital CHComm applications [27]. For the digital multimedia application on the Menorah collection at the Hecht Museum, we utilised an eight-stage approach to design content that includes CH values. An explanation of each step is provided next, including both its implementation and evaluation. A running example of the content design and its changes over each step accompanies the description highlighting the content creation process for the 'Glass Jar' artefact under the Menorah collection.

- **Stage 0—Collection of CH values**
  Inclusion of CH values in the content for communication can only be done by finding and recording a broad selection of CH information and its associated values with the heritage in question. This is a necessary step for the entire process to be initiated. Multiple CH values are associated with the Menorah collection and the Hecht Museum. Some of these values, which are part of the overall museum, include Zionism, religious values of Judaism, the historical values associated with various heritage properties and the social/symbolic value that the Menorah symbol represents in today's world. In the case of the Menorah collection the religious values and the administrative importance of the menorah symbol comes to the forefront. The collection process uncovered a few interesting details such as the fact that the nature of values associated with the Menorah symbol changed over time.

  At one point in history, after the destruction of the second Jewish temple in Jerusalem, the Menorah was taken away by Roman conquerors as the spoils of war. This was depicted on the Arch of Titus in Rome built in 81 CE. This depiction may have led to the association of 'defeat' to the symbol of Menorah from the Jewish perspective of that time which was likely why the symbol was not depicted on the Jewish coins minted during the Bar Kokhba revolt of 132–136 CE. On the other hand, today, the Menorah is the official symbol for the State of Israel. This phase of the collection of CH values associated with the Menorah symbol itself and the artefacts within the collection was done by going through various academic sources and through open discussion with both heritage experts and non-experts who shared their views and historic knowledge regarding the Menorah. An example of CH information collected for the 'Glass jar' artefact and the relevant information sources are shown in Figure 2. It is to be noted that the information regarding the artefact provided at the museum was adequate but naturally more information and potential value associations were available from other sources. This additional information was used to enhance the communication application.

**0. Collection of CH Values**

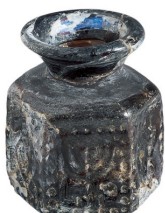

Museum Description:

Hexagonal glass vessel decorated with seven-menorah, Jerusalem 6th-7th Cent. CE. Vessels of this type are called eulogiae (=Blessings) and were manufactured in Jerusalem. They contained oil, or earth from the Temple Mount, were purchased by the pilgrims and served as Jewish funerary offerings.

| Information<br>What is the heritage? | Sources |
| --- | --- |
| Hexagonal Glass Jars with the menorah symbol were used as eulogiae (Blessings) in Jewish culture during Roman-Byzantine period. | The Met Museum, New York<br><br>https://www.metmuseum.org/art/collection/search/465957 |
| Clay vessels were also used as eulogiae | The Israel Museum, Jerusalem<br><br>https://www.imj.org.il/en/collections/200030 |
| Christian rituals also have similar eulogiae with different symbols, probably manufactured by the same craftspeople | Glass pilgrim vessels from Jerusalem: Part 1, Dan Barag, 1970 |
| Judaic and Christian eulogiae had pagan symbols decorating them | Glass pilgrim vessels from Jerusalem: Part 1, Dan Barag, 1970 |

**Figure 2.** Collection of CH information and their sources for the 'Glass jar' artefact in the Menorah collection.

- **Stage 1—Create an initial vocabulary of values**
  Information collected with regards to the CH can have multiple value associations. Since we cast a wide net to grab as many potential pieces of information and associated values, the CH information and values need to be filtered. This can be done qualitatively by the creators of the application based on previous work done by interdisciplinary researchers who have covered the heritage. This process can be supported by direct discussions and observations from experts on the heritage. By doing this, we end up with an initial set of values that can justifiably be included in the content of the digital multimedia application.

  To better categorise the information and values collected for each artefact, the provisional value typologies and their definitions suggested for assessing values in conservation planning were used [11]. This was further enhanced by deriving the 'points' of CH information and values that answer the first two out of three questions suggested by a model for significance assessment [12]. Our points answered the following questions at this stage:

  1.  What is the heritage in question?—Forms, relationships, practices etc.
  2.  Why is it valuable?—Associative, sensory, evidentiary and functional aspects

  For the Menorah collection at the Hecht museum, values relevant to the specific artefacts were filtered based on studies of works done by other researchers and ongoing discussions with experts who are working with or have previously worked with the museum. The discussions with the experts at the museum and elsewhere at this stage served to corroborate the information and their value associations. We were able to confirm that the information and values that were collected and filtered was free of errors and did not omit any pertinent historical information. The initial vocabulary of values and the points of information that contain them with regards to the 'Glass jar' artefact are shown in Figure 3.

**1. Create an Initial Vocabulary of Values**

- Categorized under typologies of The Getty Conservation Institute
- Answer the questions "What?" and "Why?"

Religious:
1. Blessings from holy sites for burials: Glass jars with menorah were used as souvenirs to bring back blessings like oil or earth from Temple mount.

2. Menorah symbol more important than material: Menorah was important than the material. Clay vessels were also used as eulogiae.

Aesthetic:
3. Specific shapes for pilgrim vessels: Glass jars were usually hexagonal or squat hexagonal (octagonal) in shape.

4. Representing Jewish identity: Motifs restricted to menorah and Jewish symbols.

5. Art Inspiration from other religions: Judaic and Christian eulogiae had pagan symbols decorating them.

Social:
6. Coexistence of Judaism and Christianity: Similar in form, colour and technique of jars with different motifs. Probably, made in the same workshop.

Stage 1
- Features of significance
  What is the heritage in question?
  (forms, relationships and practices – Stephenson 2008)

Stage 2
- Aspects of value
  Why is this heritage valuable?
  (Associative, sensory, evidentiary and functional aspects)

Stage 3
- Qualifiers of value
  How valuable is it?
  (authenticity, rarity, condition, .. etc.)

**Figure 3.** Initial vocabulary of values categorised by type and listing the information that contains them for the 'Glass Jar' artefact.

All the artefacts in the Menorah collection were noted to have religious significance and it was followed by other values that applied to each artefact. For example,

– The glass pendants were historically seen as luxury goods and this tied an economic value to the artefact at its time (2nd century CE).
– The glass jars were observed to have designs that were based on pagan symbolism, which is an information that has historical value.
– A basalt slab from the Roman-Byzantine period showed a menorah with five branches while the Israeli national emblem of the Menorah and the original menorah of the second Jewish temple is represented with seven branches. This difference in representation comes from adherence to a religious prohibition which states that seven branch menorahs must not be represented on buildings other than the Temple in Jerusalem. This is information that has religious and historic value which is also relevant to the contemporary social context as this restriction is not very strictly followed today. Many representations of the Menorah especially since the formation of the State of Israel shows the seven-branched version. This facet has social/symbolic value. The Menorah symbol with the seven-branches is inextricably linked with the identity of the State of Israel as on date.

- **Stage 2—Assess significance of values by relative importance**
  The initial vocabulary of values is now put to the test by the third question from the model for significance assessment [12].

  3.    How valuable is it?—Authenticity, Rarity, Condition etc.

  In answering this question, we ended up re-organizing our points for each artefact. Our criteria for this stage was the Authenticity (A) of the heritage information, the Rarity (R) of the artefact itself and the relevance of the artefact to the Museum context (M). Once again, this rearrangement was based on our understanding of these three aspects. The rearranged list of points covering the information and values related to the 'Glass Jar' artefact are shown in Figure 4.

**2. Assess Significance of Values by Relative Importance**

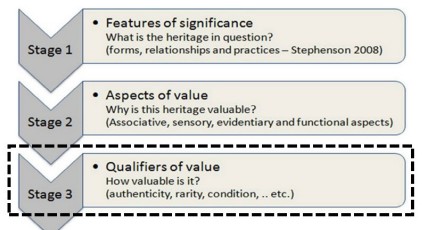

Stage 3 for identifying relative importance based on:
1. Authenticity (A)
2. Rarity (R)
3. Relevance to Museum (M)

**Stage 1** • Features of significance
What is the heritage in question?
(forms, relationships and practices – Stephenson 2008)

**Stage 2** • Aspects of value
Why is this heritage valuable?
(Associative, sensory, evidentiary and functional aspects)

**Stage 3** • Qualifiers of value
How valuable is it?
(authenticity, rarity, condition, .. etc.)

1. Blessings from holy sites for burials (A: As per literature, R: No similar artefact in Museum, M: Highlights Jewish practice)

4. Representing Jewish identity (A: Use of Menorah, shofar and lulav identifies with Jews since history , R: Not rare but important in contemporary time as National symbol, M: Museum highlights life of Jews across history)

2. Menorah symbol more important than material (A: Used across history , R: Not rare , M: Curated collection of the Museum)

3. Specific shapes for pilgrim vessels (A: Two shapes only found, R: Rare, only few pieces in the world, M: Shape not important as per Judaism)

6. Coexistence of Judaism and Christianity (A: Analysis of vessels by researchers, R: Rare because no written evidence available, M: Museum houses material culture in Eretz Israel that includes other cultures also but not primary focus )

5. Art Inspiration from other religions (A: Analysis by researchers, R: Not rare, M: Not important in museum context )

*(Importance — arrow pointing upward along right side)*

**Figure 4.** Information relating to the 'Glass Jar' artefact is rearranged based on our understanding of its significance based on the Authenticity (A) of the heritage information, the Rarity (R) of the artefact itself and the relevance of the artefact to the Museum context (M). The numbering is kept the same as in Figure 3 to highlight the changes created by this rearrangement.

After this rearrangement, a table was composed for each artefact and related concepts such as Hecht's Vision for the museum and the idea and meaning of the term 'Eretz Israel'. These tables collated the answer to the three questions that were adapted from the three-step model for significance assessment. Figures 5 and 6 show the tables made for the content design of Dr. Reuben Hecht's vision for the museum and 'Glass Jar' in the menorah collection. In doing so, some CH value associations were elevated in our assessment of their importance and some were eliminated. As can be seen in Figure 6, the association of Pagan symbology with the designs on glass pilgrim vessels was not seen as particularly important by the experts while the designers (we) chose to deliberately highlight the aspect of potential peaceful co-existence of religions from the fact that Jewish and Christian pilgrim vessels may have been made by the same manufacturers in Jerusalem.

**Reuben Hecht**

| Information
What is the heritage? | Values
Why is the heritage valuable? | Sources | Relative significance
How valuable is it? |
|---|---|---|---|
| Hecht museum was established in Haifa | Hecht believed that Haifa was the center for revival of the Jewish State | Reuben Hecht – Vision and Fulfilment By Moshe Shamir | This information is highlighted in the museum. |
| Hecht spent sixty years collecting archaeological items that represented the Eretz Israel's material culture. | Ancient relics were proof of the Jewish people's connection to Eretz Israel, and archaeology would be an expression of Zionism | Hecht Museum website | |
| Hecht used to collect coins from childhood and had funded many archaeologists and their excavation projects. | Hecht was a Jew who believed in the concept of "Eretz Israel" and connection between People and Land of Israel. | Reuben Hecht – Vision and Fulfilment By Moshe Shamir | |

**Figure 5.** Content design structure showing the relative importance of the information and values associated with Dr. Reuben Hecht's vision.

**Glass Jars** 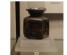

| Information<br>What is the heritage? | Values<br>Why is the heritage valuable? | Sources | Relative significance<br>How valuable is it? |
|---|---|---|---|
| Glass vessel with the menorah symbol were used as eulogiae (Blessings) in Jewish culture | Religious significance - pilgrims when they visited the holy sites was to take something of the blessing of the site back home with them, to serve as a remedy or source of protection. Such "souvenirs" often consisted of oil from the lamps that burned at the holy sites, water from the Jordan River, or earth from a place associated with a holy person. | The Israel Museum, Jerusalem | This information is highlighted in the museum.<br><br>The contemporary equivalent of this practice (if any) can be highlighted |
| Clay vessels with the menorah symbol were also used as eulogiae | The menorah symbolism is the point of significance, and the material of the vessel is not as important. The belief is anchored by the symbology | | This explains how and why the menorah symbol is significant and why the glass jar is worth preserving |
| Christian rituals also have similar eulogiae with different symbols, probably manufactured by the same craftspeople | There may have been a historic period when both the religions peacefully coexisted, and their rituals were practiced side-by-side | Glass Pilgrim Vessels in Jerusalem: Part I by Dan Barag | This is an important point that can be highlighted as a call out to religious harmony. May be more relevant in other contexts. |
| Judaic and Christian eulogiae had pagan symbols decorating them | Common ancestry and pagan roots were carried over in the religious practices | | This may not be as important in the current context |

**Figure 6.** Content design structure showing the relative importance of the information and values associated with the 'Glass Jar' artefact in the Menorah collection.

- **Stage 3—Dialogic Inquiry for the importance of values**
  The values that have been selected thus far and arranged based on their relative importance rely primarily on the understanding of the designers, shaped by expert views. For the dialogic inquiry stage, preliminary content is compiled and an understanding of the user perspective along with the expert views on the CH values is obtained. The tables for each artefact, like those in Figures 5 and 6, show the basic content to be included. This content can be presented using any digital platform or technology of choice.

  A set of random potential users were selected and asked about their perspectives regarding the Menorah as a symbol and an assessment of their awareness of these artefacts was also conducted. The participants of this pilot process of dialogic inquiry were selected by convenience sampling and covered people of diverse nationalities with varying levels of familiarity with Israeli culture and history. We developed videos which compiled the content from the previous step with voice-over narration and had relevant images from the museum. Potential users were asked to respond to a few questions after a video was shared via relevant pre-existing Facebook groups. In the responses, some users with knowledge of Jewish heritage showed near complete understanding of the Menorah symbol and an awareness of its related artefacts while others showed clear appreciation of the CH values included in the design. A sample of the responses to four questions for the content shared on the 'Glass Jar' artefact received via Facebook is shown in Figure 7. The four questions asked were:

  1. Which piece of content related to the 'Glass Jar' interested you the most? Why?
  2. Are you familiar with the practice of bringing back Eulogiae (Blessings) from religious sites?
  3. What are your thoughts on ritualistic purity of material?
  4. What are your thoughts on the same crafts-person creating pilgrim vessels for different religions?

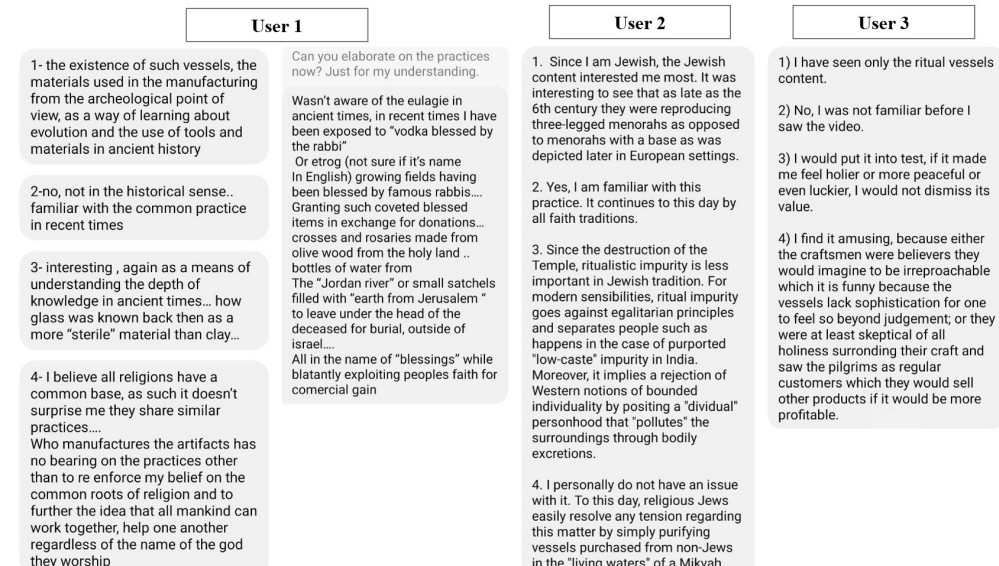

**Figure 7.** A sample of responses to four questions for the content shared on the Glass Jar artefact received via Facebook.

A similar line of inquiry was pursued with some experts and usual visitors of the Museum who were aware of the artefacts and its history. This was done to further enhance the content design to the best possible extent. This group responded to three questions instead of four and a sample of these responses are shown in Figure 8.

**3. Dialogic Inquiry for the Importance of Values**

Museum Description:
Hexagonal glass vessel decorated with seven-menorah, Jerusalem 6th-7th Cent. CE. Vessels of this type are called eulogiae (=Blessings) and were manufactured in Jerusalem. They contained oil, or earth from the Temple Mount, were purchased by the pilgrims and served as Jewish funerary offerings.

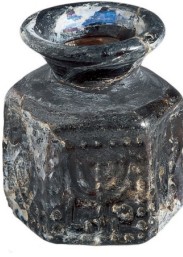

Q1. Does this artefact interest you? Why?

Q2. What more do you want to know about this artefact? Why?

Q3. Do you know about pilgrim vessels or funerary offerings?

Senior guide at the Museum and Event coordinator: (Q2) Hexagonal shapes were common for Roman jars, maybe the inspiration. No relevance in Judaism. Highlight "Why is it part of this collection?".

Museum Guide: (Q2) Stone and glass were considered pure for rituals by Jews because they don't react. Menorah makes it Jewish.

Graduate Student, Sociology: (Q3) Every year Jews visit the graves of the ancestors on religious festivals, generally carry stones and keep on the grave.

PhD student, Computer Science: (Q3) Muslims also have a similar tradition of carrying water from Mecca back as blessings.

**Figure 8.** A sample of responses from experts and knowledgeable users to three questions for the content shared on the Glass Jar artefact.

The response from the users and experts showed us that concepts such as the 'ritualistic purity' were areas with multiple perspectives. We realised that highlighting why an object was part of the Menorah collection was an aspect that the initial content design overlooked. This effort enabled us to both enhance the flow of the content presentation and gauge user interest in the heritage.

- **Stage 4—Multiple Perspectives as a Design Tool**
  The literature shows that CHComm projects have made use of multiple perspectives in their narratives. Romani (Gypsy) community memories from the World War II occupation of Czechoslovakia by Nazi Germany were included in the narrative of a serious historical game about the life at wartime. This decision reportedly received

negative reactions on online forums. The people who participated in the online discussion were citizens of Czech Republic who would potentially have an understanding of the history of Czech Republic and erstwhile Czechoslovakia. The popular discourse surrounding the events of WWII did not seem to view the Romani community or the Sudeten German community as part of the 'Czech' identity and history. This showed us that the impact of CH information and values was higher when a credible but 'non-mainstream' value association was communicated, regardless of the user perception [28].

Certain associations to religious artefacts had relevant non-Jewish perspectives apart from the Jewish religious views. In the case of the 'Glass Jar', practice of carrying Eulogiae (Blessings) from a religious site and funerary offerings is common to other religions. Users showed an interest in discussing this aspect and therefore including this perspective in the content design was necessary. The earlier choice to highlight the peaceful co-existence of religions seems to have made an impact at this point. To better illustrate this point a custom-made illustration as shown in Figure 9 was included in the content of the video with an accompanying voice-over.

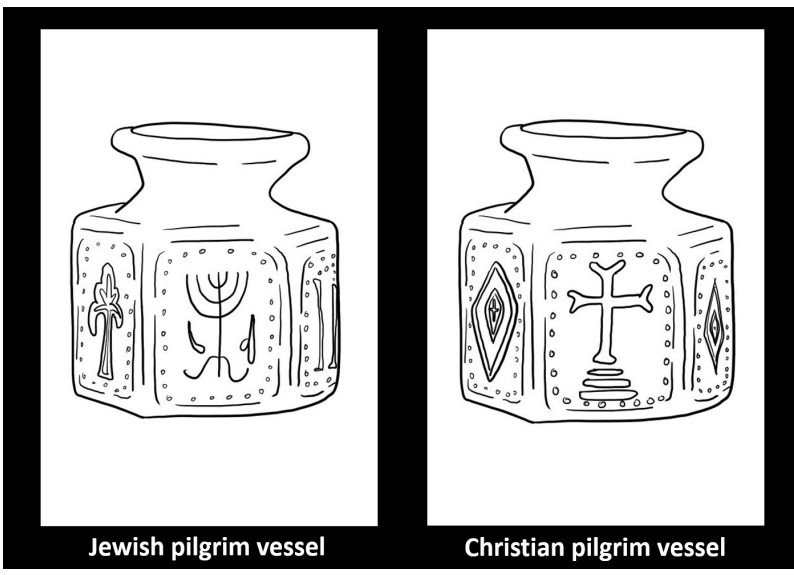

**Figure 9.** An illustration of Glass Jars with Menorah symbol and a symbol of the Cross which pointed to the possibility of the same manufacturer making the 'Glass Jars' and a potentially peaceful co-existence of both the religions at that point in time was added to include a new perspective in the content.

Inclusion of multiple perspectives necessitates mentioning artefacts and other CH properties that are not necessarily within the scope of the museum collection. This can be true for other heritage sites and institutions or collections of intangible heritage. It is advantageous for the communication of values to compare and contrast related value associations for any CH property or intangible heritage. As discussed in the serious video game example and in our experience at the Hecht Museum, different perspectives on heritage can serve to instigate user reflection and encourage discussion and further propagation of CH values. This is also a way to generate awareness for CH.

- **Stage 5—Contextualization of heritage to highlight appropriate CH values**
  Contextualization is the use of design elements or methods to communicate the basic understanding of a heritage property in its time or within its cultural context. Contextualization can be best achieved by providing 'relatable' or understandable examples. For example, in a digital CHComm project that aimed to provide audio-based narratives to augment the visitor exploration of a World War I era military camp and trenches, the designers used vivid descriptions of soldier's predicament in the trenches. These descriptions were created from journals and diaries of the time and

the 'disembodied audio' that was played in specific locations on-site helped users transport themselves into the shoes of the people from a bygone era [29]. In every case, the best approach for contextualization is to be selected by the respective designer(s). This would depend on which values can be best contextualised and which ones would need to be contextualised for a better appreciation.

In our project, to achieve better contextualization we chose to create original illustrations and change the script for the voice-over narration of our videos. In the case of the 'Glass Jar' from the Menorah collection, an illustration of the act of using Eulogiae (see Figure 10) and explanatory voice-over lines were added to the video to better communicate the meaning of the term 'Eulogiae' (see Figure 11). Contextualisation of the non-Jewish perspective was also done with an illustration as seen in Figure 9 (above).

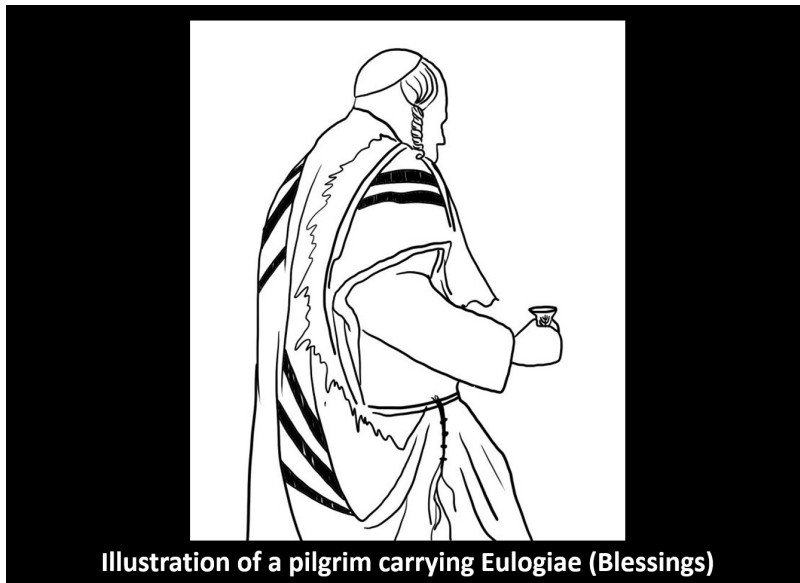

**Figure 10.** An illustration of the act of using Eulogiae created to explain the use of the Glass Jars with Menorah symbol.

Pilgrims who visited the holy sites were to take something as a blessing from the site back home , to serve as a remedy or source of protection. Such "souvenirs" often consisted of oil from the lamps that burned at the holy sites, water from the Jordan River, or earth from the Temple Mount or a place associated with a holy person.

Bringing back holy water from religious sites is an important practice even today. Many religions like Christianity, Islam, Buddhism, Hinduism used holy water as blessings or spiritual cleansing.

**Figure 11.** Lines added to the content designed for the Glass Jar artefact to better contextualise the information.

- **Stage 6—Initial prototype design for testing: Prototype 1**
  At this stage, multiple values associated with the six different artefact types and the museum as a whole have been collected and sorted through. Table 2 lists the concepts and artefacts that were deemed as necessary to create an application for the Menorah collection. It shows the value associations that we were able to identify and a categorisation of these values based on broad typologies.

**Table 2.** Concepts and artefacts that form a part of the Menorah collection and their associated values along with a broad value typology based categorisation resulting from the previous steps.

| Artefact (Tangible)/ Concept (Intangible) | Value Associations | Values Categorised |
|---|---|---|
| Dr. Reuben Hecht | 1. Haifa as the centre for revival of the new Jewish State 2. Jewish people's connection to Eretz Israel 3. Archaeology as an expression of Zionism | 1. National Identity—Social and Symbolic value 2. Communal Identity—Historic, Social and Symbolic value 3. Religious Identity—Social, Symbolic and Religious value |
| Eretz Israel | 1. Religious origins of the state of Israel 2. Ambiguously defined geographical area | 1.Religious value 2. Political motivations—Social and Economic values |
| Menorah (as the symbol and the object) | 1. As the primary Jewish symbol that has "followed the Jewish people" over the centuries. 2. As a symbol of hope, renewal and redemption 3. A seven branched candelabrum used only within the Holy Temple of the Jews in Jerusalem | 1. Communal Identity—Historic, Social and Symbolic value 2. National Identity—Social and Symbolic value 3. Historic Religious use—Historic and Religious value 4. Contemporary religious symbol—Religious and Symbolic value |
| Basalt Slab with a five-branch Menorah representation | 1. Assurance of Jewish existence in the region 2. Respecting the Talmudic prohibition of seven branch menorah representation outside the Jewish temple 3. Ritualistic significance of the menorah, shofar and lulav shown on the slab | 1. Communal Identity—Historic, Social and Symbolic value 2. Religious Identity—Social, Symbolic and Religious value 3. Religious use—Historic and Religious value 4. Architectural feature—Aesthetic value |
| Oil lamp, Lamp handle with seven-branch Menorah and Handle mould | 1.Religious significance, rituals and practices 2. Change in religious views over time—Synagogues begin to be seen as 'mini-temples' and the seven-branch menorah is represented outside the temple | 1. Religious use—Historic and Religious value 2. Historic manufacturing—Historic value 3. Change of value association—Social, symbolic and religious value |
| Token, Stamps and Seals, Coins | 1. Period of Jewish dominance of the region as shown in trade and commerce 2. Understanding of hierarchy and historical record 3. Religious adherence even when the Jewish dominance was fading 4. Symbol of the community to mark the communal produce | 1. Historic dominance—Historic and Social value 2. Record of the regions history—Historic and Social value 3. Historic Religious use—Historic and Religious 4. Social and symbolic value |
| Glass jars | 1. Pilgrims wish to take a blessing from the holy site 2. Peaceful co-existence of religions 3. Design adopted from Pagan religions | 1. Historic Religious use—Historic and Religious value 2. Religious harmony—Social and Religious value 3. Religious and Social evolution —Social, Aesthetic and Religious value |
| Glass Pendants | 1. Expression of the messianic hopes for the rebuilding of the Temple in Jerusalem 2. Non-perishable luxuries that accumulate in administrative centres like Jerusalem points to a period of Jewish socio-economic dominance | 1. Religious hopes and symbol of belief—Religious value 2. Communal history—Social and Economic values 3. Artistic expression—Aesthetic value 4. Historic social order—Historic and Social value 5. Historic manufacturing—Historic value |

Two sets of videos were created for all the artefacts highlighted in Figure 1 and this was considered as Prototype 1. One set of videos would contain the heritage information and values that the user can receive from a thorough examination of all the panels provided within the museum. This set of videos were termed as the 'CHComm Info' or info version application. Another set of videos included more information and

values that were selected by us and refined through expert and user interactions discussed in the previous steps. This was termed as the 'CHComm Info+Values' or values version application. The Info version did contain some extra information, illustrations and included some CH values that may not be easily available only within the museum premises since that was seen as aiding the flow of the presentation and also necessary to a cohesive understanding of the details. The Values version had even more content and custom-made illustrations presented as succinctly as we could. A comparative slide showing the differences in the script for voice-over and the additional illustrations for the 'Glass Jar' artefact is shown in Figure 12. The reason for creating two versions of the digital application was to test for differences in CH value appreciation that may be perceived in the visitors of the museum who chose to view the videos. The 'CHComm Info' version was essentially the control group and the 'CHComm Info+Values' version was our experimental group.

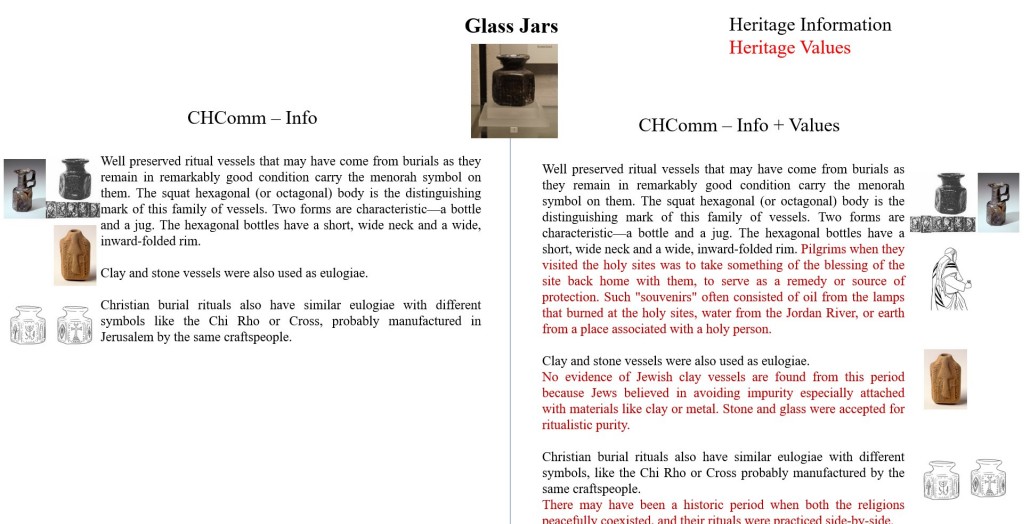

**Figure 12.** Comparative slide showing the differences between the script for voice-over and the additional illustrations in the videos made as part of the CHComm Info and the CHComm Info + Values applications for the 'Glass Jar' artefact.

The details on Dr. Reuben Hecht's vision for the museum and the explanation of the concept of 'Eretz Israel' were combined into one video while another video explained both the artefact that is the original Menorah and a timeline showing how the Menorah symbol came to be an important Jewish symbol. These videos also contained relevant images to represent the aspects being discussed and had custom-made illustrations that contextualised the information. A web-based application which allowed users to explore videos related to the Menorah artefact collection was created following the highlighted structure shown in Figure 1. Videos for individual artefacts were all kept to under 3 min in length with most being only 1 min and 30 s long. A screenshot of the web-based application with the introductory video is shown in Figure 13. Integrating the values and the information satisfactorily meant that the values version ended up being longer than the info version of the application. For instance, the info version of the content covering the 'Glass Jar' artefact was 58 s long while the values version was 1 min and 58 s long. This difference in length may appear problematic at first glance but it was necessary to our approach. A laser-focus on the delivery of information that answers "What is the heritage in question?" has overshadowed the need to explain the significance of the heritage. Answers to the questions "Why is the (particular) heritage valuable?" and "How valuable is it?" are known to the experts in the field but are not well-communicated in the current museum environment. Results from a study on a digital CHComm application that explained the design of a traditional Malay house has shown that users would like to understand the

'significance and the value of traditional Malay house, mostly focusing on the "why" and the "how" type of information rather than simply telling them the descriptive nature' of the heritage [26].

**Prototype 1: Multimedia Application**

**Presentation and Interface**

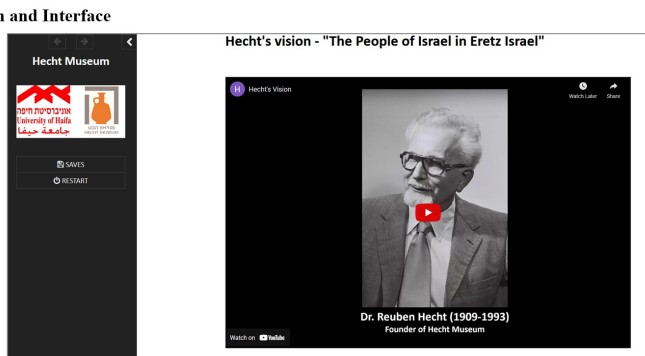

**Figure 13.** A screenshot of the web-based application with the introductory video.

A reductionist approach to the delivery of information appears to be a self-reinforcing cycle where the value associations of a heritage are not perceived as 'core' in the communication about a heritage and get left out. A museum guide who reviewed the content remarked that they were surprised by the amount of information collected and wanted to receive the sources of information and values that were to be communicated in our application. These sources were to be added to the museum's repository and as such this highlights the importance of values in any effort at communication of heritage. The additional text and illustrations in this case was purely to contextualise the value associations while providing differing and authentic perspectives on the heritage where necessary. As laid out in the stages previously a deliberate and meticulous refining of the content was pursued to include all the information and values with their significance and nuances. The variation in content length was the inevitable result.

- **Stage 7—Targeted evaluation via User Feedback**
  Visitors who had gone through the menorah collection displayed at the museum were asked if they would like to learn more about the artefacts by watching videos. This was done at different times of the day over the period of a week and fourteen visitors used the web-based application to watch relevant videos about the Menorah collection. Seven of these users were shown the Info version of the application and the other seven were shown the Values version. Visitors were not aware of the existence of a different version and both groups were asked a similar set of questions. No single user was expected to watch videos related to every single artefact under the Menorah collection. Most users watched a video related to one artefact, while some users chose to view videos for more than one artefact. At least five questions were asked to all users and one or two additional questions were asked depending on their responses. The first three questions, which were common to all users, assessed their appreciation for the vision of Dr. Hecht, the concept of 'Eretz Israel' and the history behind the Menorah as an artefact and a symbol. The last two questions were based on the specific artefact video viewed by the user. In the case of the values version of the 'Glass Jar' artefact, the last two questions tried to see if the user was familiar with the concept of Eulogiae and how they felt about the fact that the same manufacturer may have created jars for both Christian and Jewish pilgrims. Some users were also asked for their thoughts on the idea of the 'ritualistic purity of certain materials' in case they declared that they were of Jewish heritage (cultural insiders).

The responses showed that users were happy to have been provided with the content and many were eager to discuss their views on certain aspects of heritage. Users with a Jewish heritage had no issues interpreting the content while some non-Jewish and non-Israeli visitors seemed to struggle to understand the values behind the symbolic representation of the Menorah. One user with a Jewish heritage who viewed the values version stated that they understood how the 'bronze menorah' lamp handle would have been part of a clay oil lamp after viewing the video on the 'Lamps' artefact. Another user who was not a Jew or an Israeli national stated that they understood why the Menorah symbol carved on the 'Basalt Slab' artefact only had five branches instead of the usual seven after watching the video. They had mistakenly assumed that this depiction was a 'barbarization' (an erroneous depiction) when they saw the physical artefact but the video which explained its significance cleared their understanding. This clearly showed us that our intention of including values was enhancing the CH communication to visitors. This same user also remarked that they could not appreciate how the Menorah symbol was associated with 'rebirth'. This pointed us to the fact that we might need to add further illustrations and alter the script explaining this particular association within the 'CHComm Info+Values' version. Another non-Israeli and non-Jewish user who viewed the values version and identified themselves as a non-religious person, stated the following after viewing the application:

> This area always belongs to Israeli people (sic). I can't say because I haven't been to Palestine, so I don't know the truth. Maybe because of the content I can say it belongs to Israel.

This statement echoed that particular user's perception of the contemporary socio-political realities of the region, but the Prototype 1 of the application had not explicitly commented on this. The museum is themed around the 'Land and the people of Eretz Israel' and the selected target group of artefacts covered by the application was the Menorah collection which is a primary Jewish symbol and the emblem of the modern state of Israel. This shaped the design of the application thus far and would have influenced the conclusions arrived at by this particular user who is very much an outsider to the cultural context. As a result of this response, we decided to include a clear outline of the history of Israel and where the collections within the museum stand on the historic timeline. It was necessary to explicitly acknowledge the historic and contemporary spatio-temporal boundaries of Israel as a part of the presentation in the application. The changes made to Prototype 2 as a result of this has been discussed in detail further along as a part of our takeaways.

- **Stage 8—Iterative design: Prototype 2**
  The prototype developed and evaluated using the previous seven stages was bound to have shortcomings and an improved iteration was necessary before a full assessment. This was because we expected that there would be some blind-spots in the initial design that had been instantly made apparent by user evaluation. The targeted evaluation of Prototype 1 showed that users appreciated the content but two fairly predictable issues popped out.

  1. The values associated with the Menorah were not easily appreciated by non-Jewish visitors. This showed us that improvements were needed for this part of the content. Especially the contextualization and the explanation of religious perspectives needed some reinforcements.
  2. People who watched the longer bits of video noted that the content needed to be shortened. This led to a decision to split and rearrange videos such that the longest of them would not be more than 2 min.

We observed that the CH values included were appreciated for all the artefacts by the users. The responses showed that users understood the implications of the information better than they would have by only looking at the artefacts in the museum. Many users felt that the content shown enhanced their knowledge and even the users who

knew about most of what was discussed in the videos were appreciative of what they saw.

A redesign of the content implied that the evaluation questionnaire needed to be re-written since some of the videos were shortened and/or split to reduce their length. The revamped version of the application which is the Prototype 2 is also the final version that will be analysed in this paper. This final version can be accessed online at the following links:

- CHComm Info version:
  https://universityofhaifahecht.on.drv.tw/I2HechtMuseum.html (accessed on 25 July 2022).
- CHComm Info + Values version:
  https://universityofhaifahecht.on.drv.tw/V2HechtMuseum.html (accessed on 25 July 2022).

We also decided that an improvement to the UI of the application was warranted. It was the aspect that was least focused upon in the first prototype of the application. Once the redesigned content was finalised and re-recorded with an improved voice-over narration and a new questionnaire was created, a final round of assessment was conducted.

The new questionnaire design had six total questions. The first two open-ended questions covered the users perception of the Menorah symbol and their understanding of Dr. Hecht's vision. The next two questions were directed at finding out why the users chose to view the video on a specific artefact among others and what interested them in the content. Further, they were also asked if they were already aware of the content shared. Based on the responses to these questions a few more follow-up questions were asked to assess if the user found the content worth their time. The fifth and sixth questions were specific content-based questions and depended on the video viewed by the user. The fifth question aimed to understand the extent of user perception of the value(s) included in the video. The expectation was that the users of the values version may have more to say about the artefacts while the users of info version may have further questions which they would like answers for. Most artefacts in the collection had a contemporary use or implication. For example, jewellery is still a luxury contemporary-use item, there are lamps used for religious purposes and food products that conform to religious stipulations. Therefore the sixth question was a contemporary-use based question intended to encourage user reflection on the artefacts, heritage and its meanings both in the historical and the current contexts.

Experimental Setup

Two tablets pre-loaded with the application were used for the experiment. One had the Info version and the other had the Info + Values version. Visitors willing to participate in the study after their exploration of the menorah collection in the museum were given tablets with earphones and briefed on viewing videos. The participants were asked to specifically watch three videos and at least one or more videos covering the artefacts in the collection as per their choice. Each survey took 15–20 min per participant which included a maximum of 10 min for going through the application and watching videos and 5 min to answer the evaluation questionnaire.

There were instances where some visitors viewed the videos but did not wish to answer the questions and some visitors who did not fully view any artefact videos but still chose to respond to the survey. A total of 25 visitors viewed the entirety of at least one artefact related video apart from the videos covering the background on Dr. Hecht, the museum and the Menorah symbol. The Info version of the Prototype 2 was viewed by 12 participants and the Info + values version was viewed by the other 13 participants. No participant was aware of the existence of another version and everyone received the same set of questions that were prepared based on the previously explained philosophy. The process of asking questions to the user and their responses were recorded as audio files with

their knowledge and permission. Certain demographic identifiers and a self-assessment of religious inclinations were collected but no identifiers of personal details were recorded. The results from the analysis of these open-ended responses and a discussion of the same is covered in the following section.

## 4. Analysis and Discussion

All participants in the evaluation were educated and reported as having completed high school education. Every participant was also conversant in the English language which was necessary to understand the videos. Even so, there were only three native speakers of English out of the 25 total participants. The open-ended responses to the six questions were recorded and transcribed. The text files generated from this transcription were then processed using the AntConc version 4.1.3 which is a freeware that can be used for concordance and text analysis of a corpus of text files. This method was used to obtain word counts and conduct pattern finding studies in the responses. Another aspect of the analysis was a qualitative assessment that was designed as part of the targeted evaluation questionnaire. The responses to the specific understanding and content appreciation based questions were analysed in the context of the visitor demographics. The responses were also studied for signs of perception of values that were included or a lack thereof in case the user viewed the Info version of the application.

### 4.1. Summary of Analysis

All participants commented that the Menorah was an important religious and national symbol according to their understanding and users who were also Israeli nationals felt that their perspective was included in the content. Users of both versions commented that they were able to understand Dr. Hecht's vision better from the application than from the panels explaining the same displayed on the walls. Users of the Info + Values version stated that it was clear how the land and the people of Israel were represented by the various collections at the museum (not just the Menorah collection) after viewing the content. They also stated that the discovery of these artefacts and the understanding of the evolution of the Menorah symbol over time confirmed their view that Jews lived on this land long ago. Some users felt that this helped them reinforce a belief that Israel was a land of Jews in ancient history. The belief that they carried was substantiated with the knowledge. A number of users, who identified themselves as Jewish, shared their perception of Menorah by quoting religious teachings and ideologies with anecdotal stories, for example, a user translated a Hebrew saying as part of their response "If you light a candle with my light, we both have shared the light".

Participants who viewed the Info + Values version and identified themselves as believers of Judaism responded that they knew the information and values shared about the Menorah and felt that the content could be used to teach other people about the subject. A user appreciated the application and praised its authentic and credible knowledge-sharing. Two users with Jewish heritage commented that they were interested in knowing more about the stamps and were able to learn a new point about their historic use. The stamps were used for marking the special 'matzah' bread made during Jewish religious holiday of Passover by Jewish bakers and they had the symbol of the seven-branch Menorah on them. The description in the museum beneath the stamp states that it belonged to a Jewish baker in the Roman-Byzantine period. This does not explain how the stamp was used to mark the ritualistic purity of the bread for consumption by Jews. The stamp is shown and the 'description' of what it is has been provided but an 'explanation' of how it was used or the 'significance' of why it was important to a community in a period in history is not provided. Users pointing this out as a new learning showed us that the deliberate value inclusion was being appreciated.

Participants who viewed either version and were not followers of Judaism commented that there were symbols or representations that were of a similar importance as the Menorah is to Judaism in their religious belief systems such as the Cross for Roman Catholics or the

Crescent moon for Islam or the symbol of Om for Hinduism. It is of particular interest here that the concept of the Jewish Temple in Jerusalem being a central focus of Judaism was lost on some non-Jew users. When the difference was explained to them, they stated that they had missed the point. Some visitors were unable to understand how or why Menorah was seen as a symbol of redemption or renewal. This could be related to the previous observation of non-Jewish users not being able to differentiate between the Jewish Temple and a Synagogue. In contrast, self-identified believers of Judaism spoke about the representations of Menorah in a synagogue as being inside or outside the 'Temple' without any confusion. Visitors who were not citizens of Israel and did not have Jewish heritage reported that they had not understood why the Menorah symbol was on many places (such as the Israeli passport) or that they had not appreciated why the Menorah was seen as so important. The content had helped them understand and appreciate its importance.

One user of the Info + Values version felt that the museum on its own was not capable of making people without a prior knowledge of Judaism understand the meanings behind the artefacts. This user also commented that the application can help at some level but the emotional impact of the information included in the content would be lesser on a non-Jew especially when compared to a Jew. The implication here seems to be that those with a Jewish heritage may be emotionally invested in the artefacts and their history regardless of their level of religious belief or knowledge of history. The additional information and values provided by the application does add value and meaning to their experience also. Commenting on the specific content and their content choices within the application, two users reported that they chose to view videos on the stamps/tokens because they felt that it looked like jewellery and they like jewellery. Another user reflected on contemporary uses of the artefacts with a comment that seeing factory-made (contemporary) lamps in the shape of religious symbols such as the Star of David or the Menorah was important to them since it was a way of preserving their history and tradition. A user who said that they were well aware of the artefacts respective histories stated that:

> Basalt is a very hard stone to work. The stone is far away from Jerusalem and only available in Golan heights. The ancient community had the motivation to bring the stone, had the wealth to hire a craftsman who would put in the effort to carve the stone.

This, according to the user, communicates more of the values that were placed by the ancient community on its religious buildings and symbols. One user who used the Info version of the application wondered why the Menorah representation was only on coins from a certain period. Another user who used the value version of the application appreciated learning that the Menorah representation was at one point seen as a symbol of defeat which is why the coins from 130 CE did not have a Menorah representation.

The responses fell mainly within the expected patterns and the users who viewed the Info + Values version of the application responded with more clarity and appreciation for the CH values included in the application. An unexpected outcome was the failure to communicate the meaning of the Jewish Temple in Jerusalem in Judaism and its association with the Menorah symbol. On the one hand this was never explicitly covered by any of the artefacts in the museum and on the other hand we had not prominently presented this information.

A 4-word limit N-Gram count of the responses showed that out of the 12 respondents who viewed the Info version, 4 participants stated that they did not know something, while three stated that they were able to learn 'a little bit more [than what they already knew]' from the content. This shows that about half of the participants were aware of the content displayed in the Info version of the application. The majority of the information within the Info version was sourced from artefacts and text panels within the museum and yet 50% of the viewers of the Info version were able to learn something new. This could be due to the fact that the application reinforced or highlighted some information and associated values from the content presented at the museum. This shows that even a digital multimedia reiteration of the existing information can benefit a casual visitor.

The same counting method was applied to the responses for the Info + Values version of the application. The analysis revealed that 70% of the users stated that they did not know something they saw in the content. This meant that more than two-thirds of the users in this case were able to glean new information by viewing the value added version of the application. Looking at this from the other side, three users each of the Info + Values and Info versions of the application stated that they knew about or were already aware of the content they watched.

'I am not sure' is another often repeated 4-word phrase that appeared three times and all from the same user of the Info version group and six times from three different respondents of the Info + Values version. Looking at the contexts within the responses, to understand why participants were 'not sure' revealed that users attempted to process and reflect upon new information that they received, which left them unsure about their interpretations. The videos were presented to the participants at or near the display for the menorah collection within the museum and the questionnaires for both versions were administered as soon as the users watched the videos. This could be one of the reasons why the users found it difficult to consolidate their thoughts. Another aspect is also that the language for the questionnaire and discussion was in English, which is not the local language of Israel or the native language for most of the participants who viewed the videos.

The analysis of the word 'interesting' or its variations, using a keyword in context (KWIC) search, showed that seven respondents for each version had found something that interested them. This would mean that a marginally higher number of the Info version respondents were engaged by the content than the Info + Values version. This is not surprising since both versions included custom-made illustrations and well-edited voice overs with a script that was refined after feedback from the testing of Prototype 1. Even so, the overwhelmingly one sided effect with regards to new knowledge delivery was not seen here. A similar result came for the words 'emotion' and 'meaning' which appeared once each from users of both versions.

Using the same search method for the word 'important' returned a ratio that was more in favour of the Info version of the application. Once again, seven out of 12 respondents of the Info version had felt that some point in the content was important but only five out of 13 of the users of the Info + Values version used this word. This could be attributed to what ended up being a shorter viewing time for the Info version when compared to the Info + Value version purely due to the fact that the former had less content and therefore ended up enhancing the impact of what was already there.

*4.2. Discussion and Takeaways*

Both versions of the application undoubtedly provided knowledge and generated interest among the users even if they were previously aware of the information. Figure 14 shows the result of a word cloud generated from the 4-word limit N-Gram analysis done with the responses from both versions to the questionnaires of Prototype 2. A 4-word limit N-Gram was chosen over a higher or lower word N-Gram since 3-word or lesser N-Grams would pick more initiating or connecting phrases such as 'something to do' or 'I think it'. This was not always a useful indicator of intention or understanding. Higher word-limits reduced the repetitions as there were very few instances of more than five words being repeated among responses and the intention of the respondent was already clear when looking at 4-word phrases. The word cloud generated with the 4-word N-Gram clearly shows variations of 'I did not know' as a big part of the responses to both versions. The Info version also has 'a little bit more' as part of its major 4-word responses.

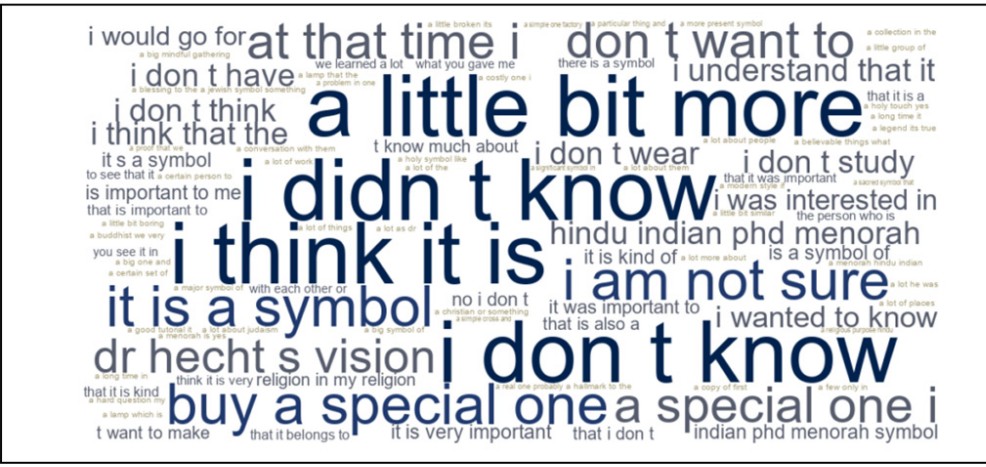

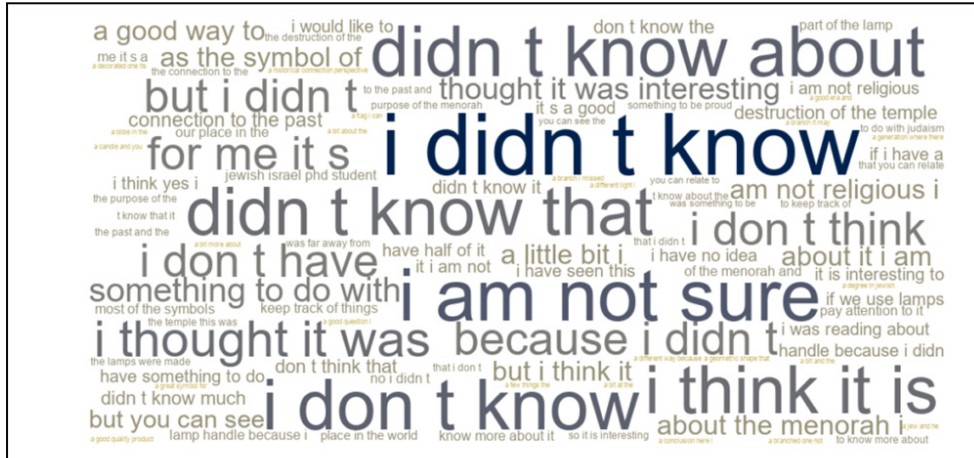

**Figure 14.** Word cloud generated by the 4−word N−Gram analysis done on the responses to the Info version (**above**) and the Info + Values version (**below**) for the Prototype 2.

We proceed to list some of the specific takeaways from this study post our analysis of the results as well as some limitations. The takeaways are split into two categories, first we list our observations that are related to the 'digital multimedia' aspect and then we list observations that are resulting from the 'deliberate inclusion of values'.

4.2.1. Points of Note for Digital CHComm Applications

1. Digital companion application—An application that condenses the information presented for each artefact collection within a museum is universally beneficial. There are bits and pieces of information both at the surface level and even deeper within the heritage and history that remain hidden from visitors who enter and look at objects in a museum. Such a companion application need not be created for every artefact collection or every heritage topic but an overall vision and reasoning behind the major collections need to be made accessible digitally. This is not a demanding task for any museum or by extension any GLAM (Galleries, Libraries, Archives and Museums) institution. This information is usually available on the walls and various other panels within the museum. Collating it with related pieces of information and making it available online to users on their mobile devices greatly enhances the communication of the CH within the museum.

2. Enhance visitor awareness and accessibility—This study was conducted only on one artefact collection at the Hecht museum out of four major collections which was once again shortlisted from 12 thematic collections and three other permanent archaeological collections. Each collection has a series of videos in English and Hebrew

previously developed and uploaded on the internet. They also have associated write-ups in English and Hebrew shown on the website apart from panels on the walls of the museum which are also in Arabic. These efforts are admirable and yet the content on the website is not always known to the visitor. Making the visitors aware of such sources of information and actively encouraging them to explore these compendiums must be a part of the visit itself. In fact, this can be seamlessly integrated with the application as mentioned in the first point. While brochures, books and other printed materials are necessary and have been widely adopted, multimedia and other digital formats for the communication of CH are also viable and necessary carriers of CHComm.

3. Glocal language—As much as possible, digital applications need to be created in local languages or in a commonly spoken and understood language for the region apart from English or other languages which are widely spoken (Spanish, French). This is necessary to achieve a smooth and easy delivery of knowledge and CH values to the user. All users who participated in the study at the Hecht museum were appreciative of the information that they received but it was evident that English, even though a common medium of communication, had its own limitations. Not every application can cover multiple languages but using at least one local language apart from a more global language is a necessity.

### 4.2.2. Points of Note from the Deliberate Inclusion of Values

4. Insider friendly communication bias—Certain CH values are well known to those who might be considered as 'cultural insiders' but this might not be the case with those who are not necessarily aware of the nuances and cultural contexts related to that community. In our study, the historical Jewish holy Temple which was and still is seen as the primary 'Temple' of the religion (existed in Jerusalem) was one such concept. Jewish believers see this Temple in Jerusalem as a central place of worship above all others. The 'First Temple' was destroyed around 586 BCE and then rebuilt as the 'Second Temple' which was then destroyed again around 70 CE. A synagogue is not a replacement for this temple and only a 'subordinate' place of worship. The hope of this central Jewish Temple being rebuilt in Jerusalem is a core tenet of Judaism. This concept being unclear to those without a Jewish heritage meant that symbolic values of 'rebirth' and other CH values attached to the Menorah symbol was not always appreciated. The artefacts and accompanying explanations within the museum and both versions of the Prototype 2 could not clarify this for some users. This means that the application is to be improved and the thematic presentation within the museum might also need to account for this. While this is one specific example for a specific case, many museums may have a version of this 'cultural insider friendly' communication bias.

5. Acknowledging boundaries—The history of the contemporary State of Israel is inextricably linked to the political and religious turmoil of the region as is the case with many other regions in the world that have constantly contested legacies. As such, the Hecht museum founded by an eventual recipient of the Israel Prize is focused on the Jewish perspective. The artefact collections within the museum focus on a period that saw the highest Jewish influence in the region which is up to the 7th century CE. The periods after this era saw a heightened Islamic influence until the 20th century. It might not be feasible to physically expand the collection at the Hecht museum or any other museum to represent every possible historic era. It is in the nature of GLAM institutions to be founded and funded with a focus on certain eras and objectives. These socio-cultural and political boundaries need to be acknowledged within the value discussions of digital applications. Contextualising the boundaries of today within their larger chronology and tracing their evolution is a worthwhile exercise to make visitors understand what is present within a museum or heritage site and sometimes what is not present. While we felt no need to court controversy, objectively

stating relevant facts and value associations by mentioning that the museum itself shows only a slice of history from a specific perspective was feasible. The first two videos covering Dr. Hecht's vision and the concept of Eretz Israel explains this with an illustrative map and a timeline shown in Figure 15. The political and social boundaries and economic constraints in the real-world need to be respected and communicated with the necessary distinction. In this case, we felt it necessary to highlight what motivated Dr. Hecht to create the museum and how the collection of the museum evolved to encompass a specific time period as a result of his motivations. This was important to acknowledge given that there are not many artefacts from the later eras in the museum as a result of the founder's vision. This helped us state clearly to the users that the series of videos shown were covering a specific period and a specific viewpoint and as such it would not be ideal to draw generalized conclusions for the contemporary socio-political overviews.

6. Engaging willing and voluntary participants—Our study even on a small subset of existing collections has shown us that there is value in exploring and deliberately including CH values in digital multimedia applications for CHComm. Museum guides being impressed by the variety of information presented and visitors appreciating the multiple value associations point to this fact. There is a pressing need to add back the 'Why?' and the 'How?' of tangible and intangible heritage into its communication apart from the 'What?'. The specifics of multiple CH properties are known academically and were not hard to access in this case but it livened up the experiences of those visitors who chose to take the time. It is worth stressing here that cutting down on the depth of the content to better serve an audience which may not be willing to assimilate even the reduced version is not advisable. The effort of a designer may be gainfully focused on engaging the willing and voluntary participants and this is where the bulk of the content design must initially be targeted. Encouraging user reflection and improving the wider accessibility to CHComm that deliberately includes values in their design is a need of the hour. When designers do not recognize the value of values, the willing visitor receives diminishing returns for their efforts. This can lead to an undesirable cycle that ends up eroding the value of cultural heritage itself.

Cultural Heritage conservation and communication go hand in hand and are rightfully rooted in the CH value perceptions of historic and contemporary individuals and societies. The value associations of any one CH property are numerous and complex and it can be a daunting task to collect, filter and present them. We followed a broad typology for the classification of values and a model of significance assessment that was described as 'purposefully simple' by its authors. The typologies and significance assessment systems we followed do not claim to be definitive and we only present them as they were conducive to our operational framework. Even so, the result of this process is a rewarding experience for willing visitors and thereby a necessary exercise for the designers of digital CHComm.

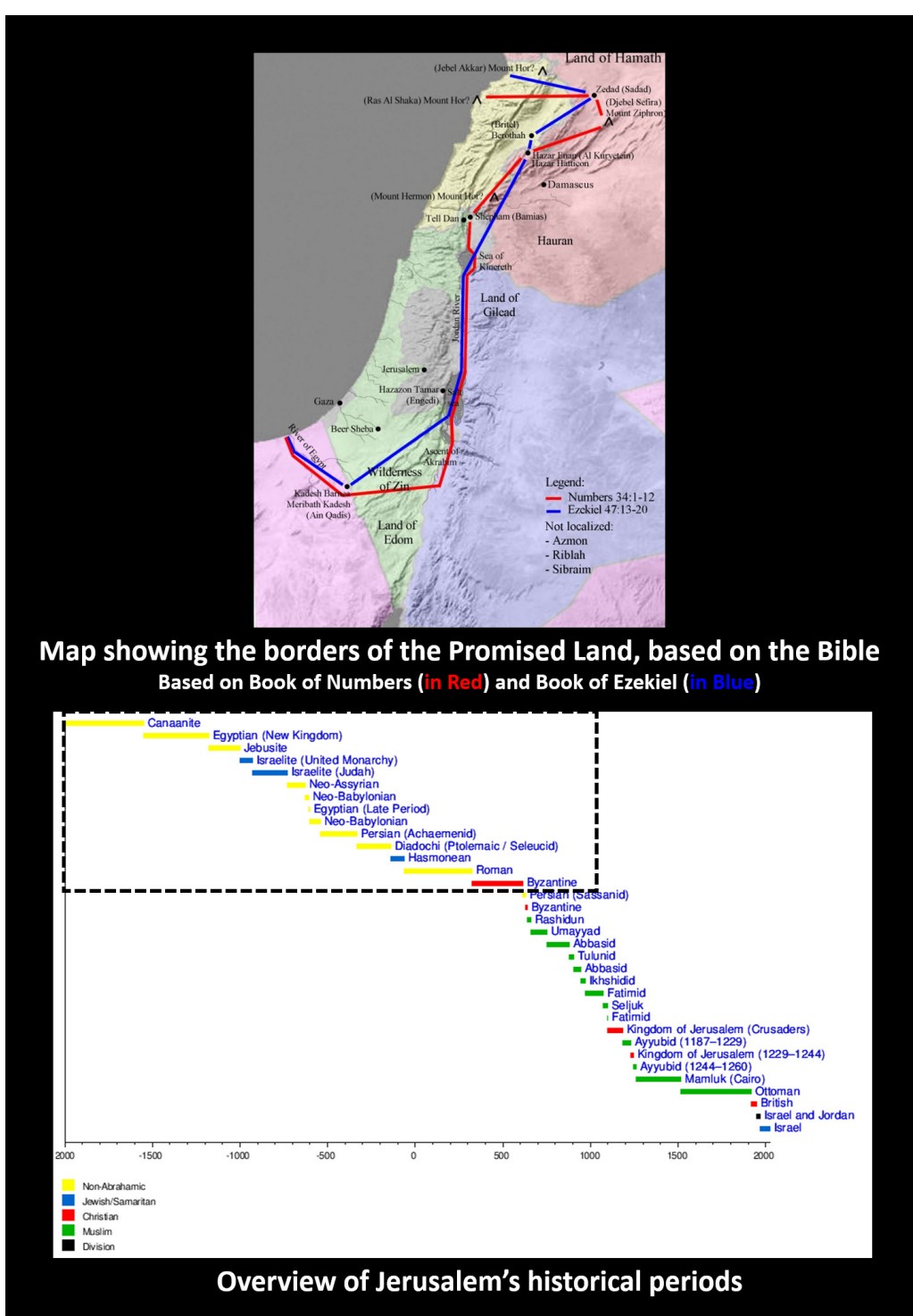

**Figure 15.** An illustrative map behind the concept of 'Eretz Yisrael' (**above**), the specific period of the artefacts in the museum highlighted on a complete timeline of the history of the contemporary state of Israel (**below**).

Two limitations in this experiment that we wish to emphasize here are that firstly the CH value associations and their typologies mentioned by us in relation to these artefacts are not the only ones that are to be considered or even an exhaustive list. The associations uncovered by our search were known to certain stakeholders but were also demonstrably 'under-communicated' to the museum visitors. These associations were verified by a peer review which included both heritage experts and non-experts and therefore can be considered as a starting point for further development. The entire study was conducted

over a period of three months and as such the time constraints made us limit our survey and collection of values to the more 'obvious' associations. We intend to expand the study to more collections and conduct a deeper analysis of value associations to enhance the quality of the content in the digital CHComm application with future iterations. Secondly, the control group that was used in this study was the users of the 'CHComm Info' version of the application. Assessing the value appreciation of the visitors to the museum who did not view any application would have provided us insights into the impact of a digital application all by itself without any intentional inclusion of CH values. As mentioned, we note that the users of the 'CHComm Info' application also stated that they learnt new information and the impact of the re-iteration of cultural knowledge by a digital application cannot be ignored even in the 'CHComm Info+Values' application. A refined and expanded study would ideally be able to control for the impact of the application being a digital summary of the artefacts that enables focused attention of the users by virtue of audio-visual engagement. Even so, as it stands our current results show the increased impact from the deliberate inclusion of CH values in a digital CHComm application.

## 5. Conclusions

This paper discussed the creation, implementation and evaluation of an eight-stage process for the inclusion of CH values and their communication to museum visitors via digital applications. The process was demonstrated by creating a digital CHComm application that deliberately included CH values built for the Menorah collection of artefacts at the Hecht Museum in Haifa, Israel. It began with the collection of associated values which were then categorised and sorted according to their perceived importance. These values were then included in a pilot application that was tested with experts and potential users after which their comments were used to refine the application and create a 'Prototype 1'. The evaluation of the prototype highlighted a couple of predictable issues apart from a few points of miscommunication or unintended communication and this led to an improved iteration with a refined presentation in 'Prototype 2'. This second version was evaluated with a redesigned questionnaire and the impact of the included values on visitors was assessed. The results point to a near universal improvement in the learning and interest shown by willing participants. Not every visitor wanted to use the application and not every visitor who used the application was ready to share their experiences but those who chose to go the extra mile felt well rewarded for their efforts. A designer might never be able to make a wide majority of the potential users actually use an application but it is very possible and desirable to create a uniquely informative and evocative application that communicates the values of cultural heritage.

Future work in this direction needs to expand the scope of the application by including more collections at the museum and test the effectiveness of such an application with local language components. This evaluation must also be done using the local language. Streamlining the content generation and value inclusion process can also be explored so that larger repositories that store the values collected and evaluation data generated can be used to further improve and fast-track the creation of such applications.

**Author Contributions:** Conceptualization, S.G., T.K. and A.W.; Methodology, S.G. and V.L.; Validation, S.G.; Formal analysis, S.G.; Investigation, S.G.; Resources, V.L., T.K. and A.W.; Writing—original draft, S.G.; Writing—review & editing, V.L., T.K. and A.W.; Supervision, V.L., T.K. and A.W. All authors have read and agreed to the published version of the manuscript.

**Funding:** This research was funded by EU Horizon 2020 research and innovation programme under the Marie Skłodowska-Curie Actions, grant agreement no. 754511.

**Institutional Review Board Statement:** Not applicable.

**Informed Consent Statement:** Informed consent was obtained from all subjects involved in the study. No Personal Identifiable Information was collected, only demographic identifiers were used for the study.

**Data Availability Statement:** The data presented in this study are available on request from the corresponding author. The data are not publicly available due to it containing multiple audio files of primary data and subsequent transcriptions.

**Acknowledgments:** This research is part of the PhD programme Tech4Culture: Technology Driven Sciences at the Università degli Studi di Torino, Italy. The authors wish to express their sincerest and heartfelt gratitude to the management and members of the Hecht Museum especially to Amir Wieliniski for his constant support and guidance throughout the project. They would also like to thank Yoav Cornfield from the Hecht Museum, Joel Lanir from the University of Haifa who provided feedback and comments which helped to improve the application and Rossana Damiano for her administrative support at the Università di Torino.

**Conflicts of Interest:** The authors declare no conflict of interest.

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
