# Peer review of "The Value of “Values”: A Case Study on the Design of Value-Inclusive Multimedia Content for the Menorah Artefact Collection at the Hecht Museum, Haifa, Israel"

_applsci, doi:10.3390/app122312330_

Round 1
Reviewer 1 Report
This paper presents the design, implementation and evaluation of digital applications using a several steps development process for cultural heritage values inclusion and their communication to the visitors. The context of the study is the Menorah collection of artefacts at the Hecht Museum in Haifa, Israel. For this, values in heritage management are first studied, then results are analyzed and discussed. The first prototype has been evaluated and improved to develop a second one and the impact of the included values on visitors was assessed
This study is not so original, as cultural heritage and digital applications have been largely explored. Nevertheless, the proposed method with the multi step process is consistent and its description is interesting for other developers for better structuring their design of digital applications for cultural heritage.
Reviewer 2 Report
This article is a well-formulated report of carefully structured research.
One tiny point: I do wonder if on page 332-33 the statement about the general public and their knowledge of the history of Czechoslovakia should read that the "did NOT have" the understanding of these conditions. At present, the test suggests the general public DID have this knowledge, but the argument really suggests they did NOT.
One larger issue. The discussion of values is clear and well laid out with good typologies of different values (aesthetic, economic, etc.). But could you possibly make a one or two sentence statement at the outset about how you define what "values" are?
Reviewer 3 Report
The theme is interesting and of extreme need when its application has real and satisfactory results for users of museums and other equipment related to cultural heritage.
Since this article is a case study, the title should be changed including in title the place where the case study was applied.
All museums or any heritage site or heritage exhibition should try to achieve universal accessibility for all, including and highlighting the accessibility of information, communication and of course also in digital format. Unfortunately, there is still a long way to go before universal accessibility is a fact. There's still a lot to do.
Of course, the heritage must be associated with valid information and that it transmits the values to memory so that it is understandable, becomes accessible and has a didactic character. Any kind of cultural heritage isolated from its context has no meaning, it becomes mute.
The literature review should be improved and the consultation of much and diverse bibliography that exists on the subject.
Study sample of interaction with visitors is greatly reduced.
This study reveals a concrete hypothesis, but many more are being done around the world so that a more in-depth study of comparison with other museums, namely related to Jewish culture, which exist even outside Israel and with associations and links to computer solutions and various types of communication with very good results should have been done.
Reviewer 4 Report
Thank you for the opportunity to review the article “The Value of "Values": A Case Study of the Design of Value-Inclusive Multimedia Content for a Museum”. The article is engaging, exploring and developing a socio-technological framework for the integration of cultural heritage values into the information content provided by a museum to its visitors.
The theoretical background is appropriate for this subject, the authors describe in the first two sections of the article some theoretical perspectives used in studying, promoting and understanding cultural heritage using digital Communication. However, I do wish the authors would consider including recent studies that have analyzed the impact of using digital technologies (digital presentation as an alternative to traditional means of displaying cultural heritage) in promoting the cultural content of museums or other types of cultural heritage.
Overall, the subject is corrected attributed to the special issue: Advanced Technologies in Digitizing Cultural Heritage of the Applied Science journal.
It is important to emphasize that the article is well-structured, and the results are clearly presented.
The proposed design with all the steps has been described well. I would have liked more details on the pretesting of the first prototype and on what sort of changes it triggered in the development of the second prototype.
Another observation that would help the authors improve the article's quality and scientific soundness is related to the experimental setup. Every time we test something in an experiment, we should have a control group in addition to the group of experimental participants. Otherwise, we can say that the participants participated in a pilot session evaluating the prototype, not in an experiment.
Also, further details of the limitations of the research could be added to the conclusions.
